# Unlearning via Sparse Representations

## Abstract

Machine *unlearning*, which involves erasing knowledge about a *forget set* from a trained model, can prove to be costly and infeasible by existing techniques. We propose a nearly compute-free zero-shot unlearning technique based on a discrete representational bottleneck. We show that the proposed technique efficiently unlearns the forget set and incurs negligible damage to the model's performance on the rest of the data set. We evaluate the proposed technique on the problem of *class unlearning* using three datasets: CIFAR-10, CIFAR-100, and LACUNA-100. We compare the proposed technique to SCRUB, a state-of-the-art approach which uses knowledge distillation for unlearning. Across all three datasets, the proposed technique performs as well as, if not better than SCRUB while incurring almost no computational cost.

## 1 Introduction

Machine Unlearning (Cao & Yang, 2015; Nguyen et al., 2022; Zhang et al., 2023; Xu et al., 2023; Kurmanji et al., 2023) may be defined as the problem of removing the influence of a subset of the data on which a model has been trained. Unlearning can be an essential component in addressing several problems encountered in deploying deep-learning-based solutions in real life. Neural networks such as Large Language Models (LLMs), which have been trained on massive amounts of commonly available data, can exhibit harmful behaviors in the form of generating misinformation, demonstrating harmful biases, or having other undesirable characteristics. A major culprit behind these behaviors is the presence of biased or corrupted instances in the training data of these models. To ensure safe model deployment, it is necessary to remove these instances. Another reason to remove instances and make a model behave as if it had not been trained on certain data is concerns about data privacy and the right of end users to expunge their data (Mantelero, 2013). For example, an individual might want their data removed from a face recognition system that was trained on their faces such that it is no longer able to identify them. All the above problems can be addressed by unlearning a specific subset of the training data, i.e., the subset of data giving rise to the harmful behavior of the model in the former cases and an individual's private data in the latter cases. Apart from these concerns, unlearning can also be used for other purposes such as removing outdated data from a model to free the network capacity for more recent or relevant data. With increasing concerns about AI safety and the increasing ubiquity of deep learning models in real-world applications, the problem of unlearning is becoming critical.

The main challenge in unlearning is maintaining the performance of the model on the data that needs to be retained, called the *retain set*, while unlearning the *forget set*. The naive way to ensure that a model has no information about the forget set is to train from scratch on the retain set. Unlearning techniques aim to achieve the same end but at a much lower computational cost compared to full retraining. Unlearning in a pretrained network is difficult, especially in densely connected neural networks, since the value of one parameter may affect the output for all the input examples given to the neural network. A possible solution is to fine tune the model we wish to unlearn only on the retain set. While this would ensure that the performance of the model on the retain set is maintained, it has been shown to be ineffective in practice in unlearning the forget set (Golatkar et al., 2020a). Other more effective solutions include retraining the model on the training data with a negative gradient for the forget set (Golatkar et al., 2020a; Kurmanji et al., 2023), or using knowledge-distillation-based training objectives to capture information about the retain set while filtering out information about the forget set (Kurmanji et al., 2023; Chundawat et al., 2023). However, all of these approaches require some form of substantial additional compute in order to facilitate unlearning. Moreover,

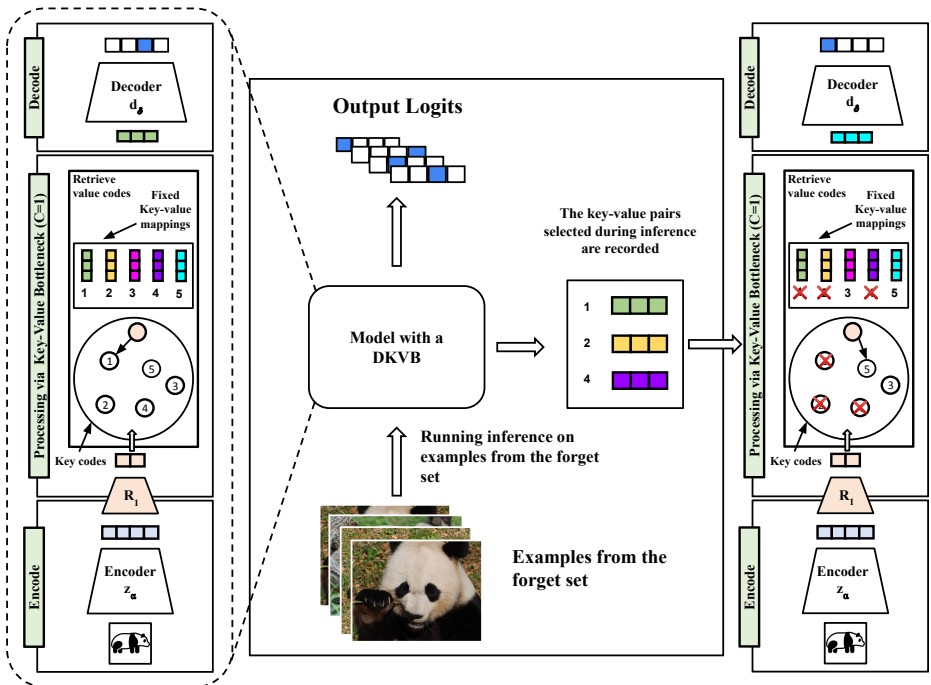

Figure 1: A summary of the proposed unlearning approach. **Left**: The structure of a key-value bottleneck. The encoder is frozen and pre-trained and $R_1$ is a random projection matrix. The values corresponding to the selected keys are retrieved to be used by the decoder. The gradient is back-propagated through the decoder into the values during training. The figure depicts the case with 1 codebook in the DKVB. However, in practice we use multiple codebooks. **Center:** Examples from the forget set are passed through the trained model and the key-value pairs selected during the forward pass are recorded. **Right:** The recorded key-value pairs are then masked from the bottleneck. As a result, the key selection is redirected to other non-masked keys, whose corresponding values are not useful for the example, and just lead to uninformed prediction.

some of the existing approaches also require access to the original training data to enable unlearning, which may not be possible in many practical applications, e.g., for a model in production which is being trained online on an incoming data stream. The use of large models is becoming more popular and prevalent with the development of general purpose transformer models. The requirement for additional compute can quickly become impractical in the context of these large models, especially in cases where a model is deployed and needs to be redeployed as quickly as possible after making the necessary changes.

In this article, we argue that *neural information bottlenecks* can be means of highly efficient and specific unlearning. Neural information bottlenecks have emerged as useful components in neural network architectures, providing numerous benefits such as improving out-of-distribution (OOD) generalization capabilities and robustness to noisy data (Goyal et al., 2021; Jaegle et al., 2021; Liu et al., 2021; 2023), facilitating large scale unsupervised pre-training and generative modeling (Esser et al., 2021; Oord et al., 2017), and more recently, helping in continual learning (Träuble et al., 2023). We particularly focus on using the **Discrete Key-Value Bottleneck** (DKVB) proposed in Träuble et al. (2023). DKVB induces sparse representations in the form of key-value pairs which are trained in a *localized and context-dependent manner*. Since these representations are sparse, we hypothesize that it is possible to remove the information about a subset of the training data without damaging the information about the rest of the data—the primary desiderata for a useful unlearning method. Moreover, since the representations are discrete, this may be achieved zero-shot, i.e., without requiring any additional compute in the form of retraining or fine tuning, by directly intervening on individual representations.

In this work, we investigate the above-mentioned idea of zero-shot unlearning in the Discrete Key-Value Bottleneck. Specifically, we focus on the problem of *class unlearning* in multi-class classification tasks, where the aim is to remove information about a specific output class, called the *forget class*, from a trained model. We use the term *retain classes* to refer to the classes other than the forget class that are present in the training data. More specifically, we wish to remove the *influence of the forget class* (or more generally speaking the *forget set*) on the model. We measure this influence using the performance of the model on held-out test datasets corresponding to the forget class and the retain classes. We propose two approaches for zero-shot unlearning on DKVB, one which requires access to training examples from the forget set—which we call *Unlearning via Examples*—and one which does not require access to training examples—called *Unlearning via Activations*. We show that the proposed methods achieve unlearning of the forget class while incurring negligible damage to the model's performance on the retain classes. We compare the proposed methods to SCRUB (Kurmanji et al., 2023), a recent state-of-the-art approach that requires additional compute to unlearn, on three datasets: CIFAR-10, CIFAR-100, and LACUNA-100, and also further investigate the effects of retraining the zero-shot unlearned models on the retain set.

## 2   RELATED WORK

The problem of unlearning has been studied in different forms for over two decades. Early works such as Tsai et al. (2014), Cauwenberghs & Poggio (2000) and Duan et al. (2007) study the problem of *decremental learning* in linear models, where a small number of samples need to be removed from a model. Ginart et al. (2019) considers unlearning as a problem of deleting individual data points from a model. They give a probabilistic definition, formalize the notion of efficient data deletion, and propose two deletion efficient learning algorithms. Guo et al. (2019) introduces *certified removal -* a theoretical guarantee of indistinguishability between a model from which data was removed and a model that never saw the data. Izzo et al. (2021) distinguishes between exact unlearning and approximate unlearning and proposes a compute-efficient approximate data deletion method, and a new metric for evaluating data deletion from these models. Golatkar et al. (2020a) and Kurmanji et al. (2023) cast unlearning into an information theoretic framework. Golatkar et al. (2020b) proposes Neural Tangent Kernel (NTK) (Jacot et al., 2018) theory-based approximation of the weights of the unlearned network. Multiple works also delve into the more philosophical, ethical, and legal aspects of unlearning and the "right to be forgotten" (Kwak et al., 2017; Villaronga et al., 2018). Chundawat et al. (2023); Tarun et al. (2023) learn error minimization and error maximization-based noise matrices which are used to finetune the trained model in order to do unlearning. Chundawat et al. (2023) further uses a generator that generates pseudo data points for unlearning in order to operate in a data-free regime.

Most relevant to our work, Kurmanji et al. (2023) introduces SCRUB, an effective knowledge distillation-based unlearning method. SCRUB considers the original model as a teacher model and trains a student model to obey the teacher model on the retain set and disobey it on the forget set. This is done by computing the KL Divergence between the output distributions of the two models and training the student model to maximize it on the forget set (this is called a *max-step*) and minimize it on the retain set (this is called a *min-step*). The student model is simultaneously also optimized for minimizing the task loss on the retain set. The training consists of *mstep max-steps*. The *max-steps* and *min-steps* are performed in an interleaved fashion.

While most of the above-mentioned approaches improve upon the naive and intractable baseline of retraining on the retain set, in terms of computational efficiency, they still require some amount of computation to do unlearning. This additional compute requirement can quickly become infeasible where large models are involved. The proposed approach, on the other hand, requires negligible computation for unlearning. Any computation that may be required is in the form of running inference on the forget set.

## 3   BACKGROUND AND NOTATIONS

**Unlearning**: Let $\mathcal{D}_{train} = \{x_i, y_i\}_{i=1}^N$ be a training dataset and $\mathcal{D}_{test}$ be the corresponding test dataset. In our experiments, we consider the setting of class unlearning, wherein we aim to unlearn a class $c$ from a model trained with a multiclass classification objective on $\mathcal{D}_{train}$. $c$ is called the

*forget class* or the *forget set*. Given $c$, we obtain $\mathcal{D}_{train}^{forget} \subset \mathcal{D}_{train}$ such that $\mathcal{D}_{train}^{forget} = \{(x, y) \in \mathcal{D}_{train} | y = c\}$. The complement of $\mathcal{D}_{train}^{forget}$ is $\mathcal{D}_{train}^{retain}$, i.e., subset of $\mathcal{D}_{train}$ that we wish to retain. Thus $\mathcal{D}_{train}^{retain} \cup \mathcal{D}_{train}^{forget} = \mathcal{D}_{train}$. Similarly, from $\mathcal{D}_{test}$, we get $\mathcal{D}_{test}^{forget} = \{(x, y) \in \mathcal{D}_{test} | y = c\}$ and its complement $\mathcal{D}_{test}^{retain}$. We refer to $\mathcal{D}_{train}^{retain}$ as the retain set training data, $\mathcal{D}_{test}^{retain}$ as the retain set test data, $\mathcal{D}_{train}^{forget}$ as the forget set training data and $\mathcal{D}_{test}^{forget}$ as the forget set test data.

**Discrete Key-Value Bottleneck**: A discrete key-value bottleneck (DKVB) (Träuble et al., 2023) consists of a discrete set of coupled key-value codes. Specifically, the bottleneck contains $C$ codebooks with each codebook containing $M$ key-value pairs. Models with DKVB use a pre-trained and frozen encoder to encode the input into a continuous representation. This input representation is then projected into $C$ lower dimension heads and each head is quantized to the $top - k$ nearest keys in the corresponding codebook. The values corresponding to the selected keys are averaged, and used for the downstream task. The keys in the codebooks are frozen and initialized to cover the input data manifold whereas the values are learnable. The mapping between the keys and values is non-parametric and frozen. Thus, the gradient is not propagated between the values and keys during training of the model. Since the values are retrieved and updated sparsely, and all the components except the value codes and the decoder are frozen, DKVB stores information in the form input-dependent, sparse and localized representations (i.e., the value codes). These inductive biases allow the framework to exhibit improved generalization under distribution shifts during training, as shown empirically in Träuble et al. (2023). Figure 1 (Left) shows an overview of a model with a DKVB where $C = 1$, $M = 5$ and top-$k = 1$.

## 4 UNLEARNING VIA SPARSE REPRESENTATIONS

**Learning a Discrete Key Value Bottleneck.** A Discrete Key Value Bottleneck (DKVB) model is first trained on the given dataset using the standard negative log-likelihood (cross-entropy loss) training objective for multi-class classification. We use a non-parametric average pooling decoder and a CLIP (Radford et al., 2021) pre-trained ViT-B/32 (Dosovitskiy et al., 2020) as the backbone in all our experiments involving the DKVB. Then we proceed to unlearn a specific subset of data from these models. Before training with the classification objective, we do a *key initialization* for the DKVB models on the same dataset.

**Key Initialization in DKVB models.** The mapping between keys and values in the discrete key-value bottleneck is non-parametric and frozen. As a result, there is no gradient (back)propagation from the values to the keys. Thus, it becomes essential for the keys to be initialized before learning the values and decoder, such that they broadly cover the feature space of the encoder. This ensures that the representations are distributed sparsely enough. As in Träuble et al. (2023), we use exponential moving average (EMA) updates (Oord et al., 2017; Razavi et al., 2019) to initialize the keys of the DKVB models. The key-initialization is done on the same train dataset $\mathcal{D}_{train}$ which we want to train the model on. The key initializations depend solely on the input encodings of the backbone and hence do not require access to any labeled data.

**Inference for Unlearning.** We propose to achieve unlearning in DKVB models by excluding key-value pairs from the bottleneck such that they cannot be selected again. This masking is done by setting the quantization distance of the selected keys to 'infinity'. Figure 1 (center and right column) shows an overview of the proposed methods. More specifically, we experiment with two methods, *Unlearning via Activations* and *Unlearning via Examples*, described as follows.

**Unlearning via Examples.** In this method, we analyze the effect of unlearning a subset of $N_e$ examples belonging to the forget set. $N_e$ examples are randomly sampled from the forget set training data ($\mathcal{D}_{train}^{retain}$) and are input into the model having a DKVB. All key-value pairs that are selected during forward propagation across the $N_e$ examples are flagged. These key-value pairs are then masked out from the bottleneck. Technically, this approach requires access to the original training data corresponding to the forget class. However, it is also possible to carry out this procedure with a proxy dataset that has been sampled from a distribution close enough to that of the forget set. This approach is motivated by the assumption that such a dataset would likely utilize roughly the same set of selected key-value pairs.

**Unlearning via Activations.** In this second method, we analyze the effect on the quality of unlearning by deactivating different numbers of key-value pairs corresponding to the forget set. We refer to the key-value pairs that have been selected as inputs to the decoder as *activations*. The entire forget set is forward-propagated through the DKVB model and all the key-value pairs selected across all examples of the forget class are recorded. Next, we mask the top-$N_a$ most frequently selected key-value pairs from the bottleneck. The requirement of accessing the original training data for this method can be avoided by caching all the activations corresponding to the forget set during the last epoch of training. Further, similar to the previous case, unlearning via activations may also be performed given access to data that has been sampled from a distribution close enough to the distribution of the forget set.

*Unlearning via Activations* and *Unlearning via Examples* are two different ways of achieving a common objective: to exclude a subset of activations corresponding to the forget set. However, using one approach over the other may be more practical or even necessary, depending on the task at hand. For selective unlearning where the goal is to forget specific examples, *Unlearning via Examples* would be necessary. In both the above approaches, we do not do any form of retraining or fine-tuning. Hence both these approaches require negligible additional compute.

## 5 EXPERIMENTS AND RESULTS

The goal of our experiments is three-fold. First, we validate that we can zero-shot unlearn information via the *Unlearning via Activations* and *Unlearning via Examples* methods in models with a DKVB (Section 5.2). Second, we show that the proposed method is competitive with SCRUB (Section 5.3). Third, we show that the proposed method is equally competitive in the less constrained setting where we allow gradient-based re-training as is commonly required by previous methods (Section 5.4). Before presenting these results, we describe our experimental setup.

### 5.1 EXPERIMENTAL SETUP

**Benchmark datasets** We validate the proposed methods using experiments across three base datasets: CIFAR-10 with 10 distinct classes, CIFAR-100 (Krizhevsky et al., 2009) with 100 distinct classes and LACUNA-100 (Golatkar et al., 2020a) with 100 distinct classes. LACUNA-100 is derived from VGG-Faces (Cao et al., 2018) by sampling 100 different celebrities and sampling 500 images per celebrity, out of which 400 are used as training data and the rest are used as test images.

**Models** On the aforementioned three datasets we study the following types of model architectures:

(a) **Backbone + Discrete Key-Value Bottleneck (Ours)**: Overall, this architecture consists of three components: 1) the frozen pre-trained backbone 2) the Discrete Key-Value Bottleneck (DKVB) and 3) a decoder, as shown in Figure 1. For the DKVB, we use 256 codebooks, with 4096 key-value pairs per codebook (approximately 1M pairs overall) following Träuble et al. (2023).

(b) **Backbone + Linear Layer (Baseline)**: As a baseline, we replace the Discrete Key Value bottleneck and the decoder in the above model architecture with a linear layer. Thus, the two components of this model are 1) a frozen pre-trained backbone and 2) a linear layer. This model will be used for the SCRUB baseline method.

In each model, we use a pre-trained frozen CLIP (Radford et al., 2021) ViT-B/32 as our encoder backbone. We refer to the appendix for additional implementation details.

**Training the Base Models** We then train both model architectures on the full training sets of each dataset. Since the backbone is frozen, for the baseline, only the weights of the linear layer are tuned during both initial training (and later unlearning). Since we use only one linear layer, we do not do any sort of pre-training (beyond the backbone), unlike in previous works (Kurmanji et al., 2023; Golatkar et al., 2020a;b). Table 1 shows the performance of these trained models on the train and test splits of the complete datasets. Starting from these base models trained on the full datasets, we will validate the ability to unlearn previously learned knowledge.

Table 1: Performance of the models on different sets of data after the initial training on the three datasets. We use two kinds of models: (a) models having a Discrete KV Bottleneck which are used for the proposed methods and (b) models where the DKVB and the decoder are replaced by a Linear Layer. These are used for the baseline. We wish to reduce the accuracy of these models on $D_{test}^{forget}$ to 0% while maintaining the accuracy on $\mathcal{D}_{test}^{retain}$.

(a) **Backbone + DKVB**

| Dataset | $\mathcal{D}_{train}$ | $\mathcal{D}_{train}^{retain}$ | $\mathcal{D}_{train}^{forget}$ | $\mathcal{D}_{test}$ | $\mathcal{D}_{test}^{retain}$ | $\mathcal{D}_{test}^{forget}$ |
|---|---|---|---|---|---|---|
| CIFAR-10 | 100% | 100% | 100% | 93.01% | 92.61% | 96.50% |
| CIFAR-100 | 99.98% | 99.98% | 100% | 78.43% | 78.24% | 96.00% |
| LACUNA-100 | 98.09% | 98.07% | 100% | 90.38% | 90.28% | 100% |

(b) **Backbone + Linear Layer**

| Dataset | $\mathcal{D}_{train}$ | $\mathcal{D}_{train}^{retain}$ | $\mathcal{D}_{train}^{forget}$ | $\mathcal{D}_{test}$ | $\mathcal{D}_{test}^{retain}$ | $\mathcal{D}_{test}^{forget}$ |
|---|---|---|---|---|---|---|
| CIFAR-10 | 93.27% | 92.82% | 97.32% | 93.02% | 92.59% | 96.90% |
| CIFAR-100 | 86.73% | 86.61% | 99.00% | 78.53% | 78.35% | 96.00% |
| LACUNA-100 | 95.58% | 95.53% | 100% | 90.68% | 90.59% | 100% |

**Unlearning**    We aim to make the problem of unlearning as challenging as possible in order to fairly evaluate the proposed methods. Therefore, on each dataset we select the class that is best learned by the respective models with the Discrete Key Value Bottleneck trained previously, to be the forget class (see Appendix A for further details). Table 1 further shows the accuracy of the base models on the train and test splits of the thus defined retain and forget class dataset splits.

**Objective & Metrics**    We report our results on the test data of retain classes and forget class, i.e. $\mathcal{D}_{test}^{retain}$ and $\mathcal{D}_{test}^{forget}$. Further, in our experiments, we aim to achieve *complete unlearning* - achieving minimal accuracy on the forget set while maintaining the performance on the retain set.[1] While achieving complete unlearning may not always be desirable, such as in the case of Membership Inference Attacks (MIAs) the proposed methods can be easily extended to defend against MIAs (We refer to Appendix D for further discussion on MIAs and the proposed methods).

## 5.2    Zero-Shot Unlearning via Discrete Key-Value Bottleneck

We will now discuss the results of unlearning via activations and examples, i.e. the two approaches proposed in Section 4 on all three benchmark datasets. All experiments are performed across 5 random seeds and mean values are reported.

**Unlearning via Activations.**    As discussed in Section 4, unlearning via activations requires us to set the hyperparameter $N_a$, reflecting the top-$N_a$ most frequently activated key-value pairs which will be masked out after inference on the forget set. We therefore start by analyzing its role over a wide range of values of $N_a$ to probe its choice and effect including $N_a = 0$ being the limit without any unlearning. Figures 2(a) - 2(c) summarize the unlearning and effect of $N_a$ on the retain vs forget test set. In the case of CIFAR-10, the initial accuracies on the retain and forget test set are 92.61% and 96.50% respectively. As $N_a$ increases, the forget class test accuracy decreases, slowly for small $N_a$ and rapidly for larger $N_a$. The model reaches random accuracy (i.e. 10% for CIFAR-10) on the forget class test data at $N_a = 150000$. At this point the retain set test accuracy is 92.97%. The model unlearns the forget class completely between $N_a = 170,000$ (0.4%) and $N_a = 180,000$ (0%). At this point, the retain set test accuracy is 92.94%, which is almost identical to the initial accuracy. Further increasing $N_a$ up to $N_a = 200,000$, i.e. about 20% of all key-value pairs, leads to an additional increase in retain set test accuracy to 93%. As can be seen from the equivalent analysis on the CIFAR-100 models in Figure 2(b) as well as the Lacuna-100 models in Figure 2(c),

---

[1] All of our experiments are performed on a RTX8000 GPU with 48GB memory.

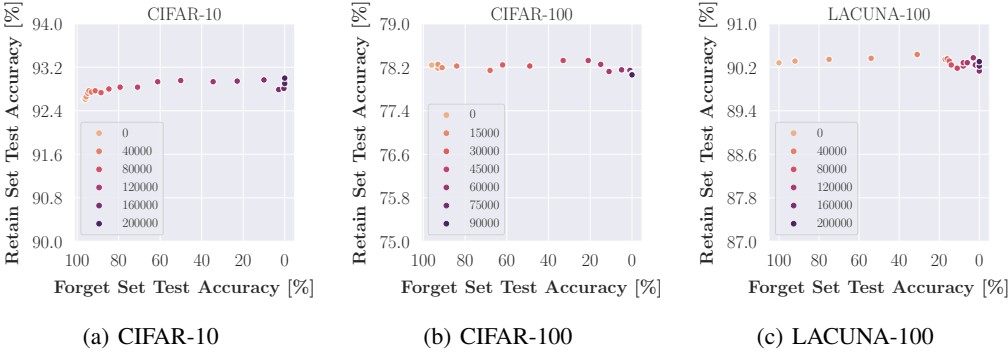

(a) CIFAR-10        (b) CIFAR-100        (c) LACUNA-100

Figure 2: Performance on the retain set test data vs. Performance on the forget set test data on (a) CIFAR-10 (b) CIFAR-100 (c) LACUNA-100 in the case of *Unlearning via Activations* as the value of $N_a$ is increased. The relative performance on the retain set test data as compared to the original models increases after unlearning in the case of CIFAR-10 and drops by 0.2% and 0.17% for CIFAR-100 and LACUNA-100.

the same trend of maintaining the initial retain accuracy while minimizing the forget accuracy up to a minimum holds across all three datasets validating its meaningful unlearning capability.

**Unlearning via Examples.** For the second method—unlearning via examples—$N_e$ examples are sampled randomly from the training data of the forget class, and subsequently used for unlearning by the mechanism described in Section 4. Similar to before, we aim to assess the effect on the choice of $N_e$ over a wide range for each dataset, including $N_e = 0$ being the limit without any unlearning. Figures 3(a) - 3(c) summarize the unlearning and effect of $N_a$ on the retain vs. forget test set. We again begin by focusing on the results with CIFAR-10 (Figure 3(a)). Here, the forget set $\mathcal{D}_{train}^{forget}$ contains 5000 examples. We start off with retain set and forget set test accuracies of 92.61% and 96.50% respectively. Similar to the previous approach – unlearning via activations – the test accuracy on the forget set decreases with increasing $N_e$. The accuracy on the retain test set, on the other hand, increases monotonically, although only slightly overall. The model achieves random accuracy on the forget class around $N_e = 2500$. The accuracy on retain set test data is at just under 93% at this transition. Finally, the accuracy on the forget set drops to 0% (i.e. complete unlearning) between $N_e = 3000$ and $N_e = 3400$ with a retain set test accuracy of just above 93% at $N_e = 3400$. Further increasing $N_e$ does not affect the retain set test accuracy notably. An equivalent analysis on the CIFAR-100 models in Figure 3(b) and the Lacuna-100 models in Figure 3(c) exhibits a very similar trend to CIFAR-10 with the only difference that the initial retain accuracy is roughly maintained instead of further increasing. Successful minimization of the forget accuracy up to a minimum is achieved across all three datasets, validating it as another option for unlearning using discrete key-value bottlenecks.

**Summary.** Both methods, *Unlearning via Activations* and *Unlearning via Examples*, successfully demonstrated unlearning of the forget class while having a negligible effect on the models' performance on the retain set. Importantly, this is achieved without any form of training, retraining, or fine-tuning as is usually required by other methods. The retain set test accuracy remains more or less constant for all three datasets except for a few minor fluctuations. This is a result of the fact that due to localized and context-dependent *sparse updates* during the initial training of the model, discrete key-representations corresponding to different classes in the dataset are well separated from each other, an important prerequisite discussed in (Träuble et al., 2023). Hence, all the information about a class can be unlearned by forgetting only a subset of the forget class training data in the case of *Unlearning via Examples*, making it very data-efficient.

### 5.3 COMPARISON WITH BASELINE

We now compare the results of both the proposed methods, which require Backbone + DKVB models against the SCRUB method, which is optimized for models without such a bottleneck.

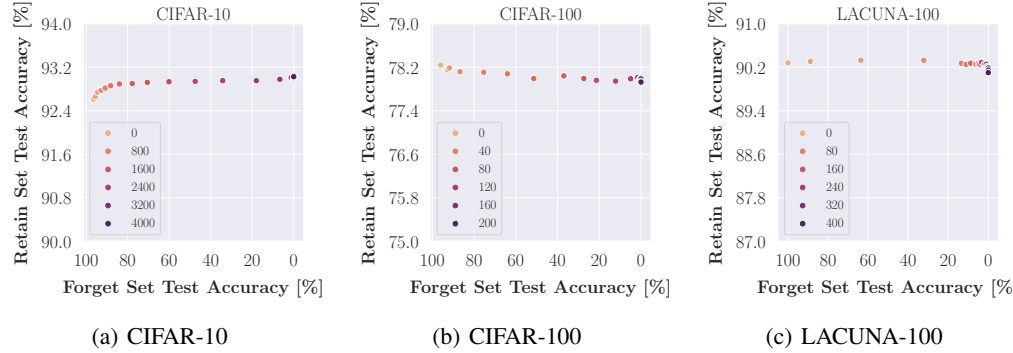

(a) CIFAR-10      (b) CIFAR-100      (c) LACUNA-100

Figure 3: Performance on the retain set test data vs. Performance on the forget set test data on (a) CIFAR-10 (b) CIFAR-100 (c) LACUNA-100 in the case of *Unlearning via Examples* as the value of $N_e$ is increased. Similar to *Unlearning via Activations*, the relative performance on the retain set test data as compared to the original models increases after unlearning in the case of CIFAR-10 and drops by 0.36% and 0.09% for CIFAR-100 and LACUNA-100.

Table 2: Comparison between the proposed methods and the baseline on CIFAR-10, CIFAR-100 and LACUNA-100. We compare the change in performance on the retain and forget test data relative to the originally trained models. All three methods are able to unlearn the forget sets completely in all three cases and maintain the performance on the retain set of the data in the case of CIFAR-10. For CIFAR-100 and LACUNA-100, the proposed methods are able to maintain the performance on the retain set better than the baseline.

| | CIFAR-10 | | CIFAR-100 | | LACUNA-100 | |
|---|---|---|---|---|---|---|
| Rel. change from base model | $\mathcal{D}_{test}^{retain}$ | $\mathcal{D}_{test}^{forget}$ | $\mathcal{D}_{test}^{retain}$ | $\mathcal{D}_{test}^{forget}$ | $\mathcal{D}_{test}^{retain}$ | $\mathcal{D}_{test}^{forget}$ |
| DKVB via Activations (sec 5.2) | 0.36% | -100% | **-0.20%** | -100% | -0.17% | -100% |
| DKVB via Examples (sec 5.2) | 0.45% | -100% | -0.36% | -100% | **-0.09%** | -100% |
| Linear Layer + SCRUB | **1.62%** | -100% | -0.91% | -100% | -1.10% | -100% |

For this, we will use the Backbone + Linear Layer model described in 5.1. On this model, we run SCRUB and compare the performance drop after unlearning with the two proposed methods.[2] Table 2 shows the comparison between the two previously reported methods and the SCRUB baseline. We can see that the proposed methods perform at least as well as SCRUB; in the case of CIFAR-100 and LACUNA-100 favorably, as we see from the damage incurred on the retain set test performance for CIFAR-100 and LACUNA-100. Finally, it is important to re-emphasize that the proposed methods achieve the shown performance without requiring any additional gradient-based training for unlearning. SCRUB is stopped when the forget set is completely unlearned without damaging the performance on the retain set. We refer to Appendix E.3 for further training details.

## 5.4 UNLEARNING BEYOND THE ZERO-SHOT SETTING

Next, we investigate the effect of using additional compute to the proposed methods. As shown previously, the proposed methods perform competitively to SCRUB zero-shot. To the best of our knowledge, SCRUB is the most competitive and relevant unlearning approach. However, it has the inherent drawback of requiring compute for unlearning. Nevertheless, for a fair comparison, we additionally explore the implications of this additional compute for the proposed two methods. Specifically, we retrain the models after zero-shot unlearning on the training data of the retain set (i.e., $\mathcal{D}_{train}^{retain}$) for 10 epochs. For the baseline, we use the same experimental setting as in Section 5.3 and run it for 10 epochs, instead of stopping once the forget set has been completely unlearned.

---

[2]We refer to Appendix E.2 for more details on the implementation of this baseline

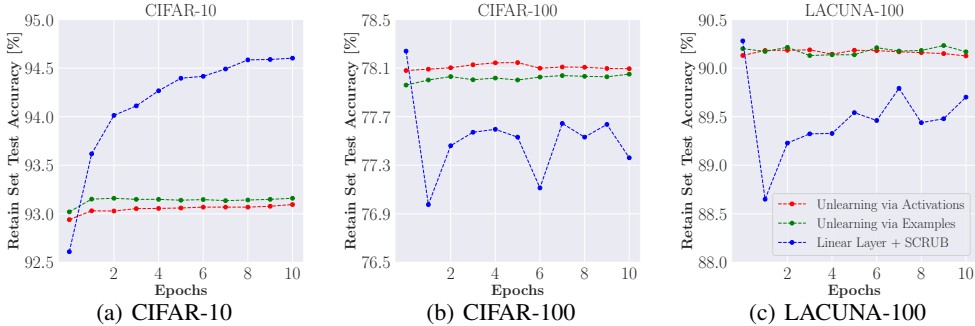

Figure 4: Comparison between the performance of proposed methods with added compute and the baseline on the retain set test data. For the proposed methods, the plots start from after the initial zero shot unlearning. For the baseline, the plots start from the original models. Retraining the models unlearned using the proposed models does not lead to any significant improvements in performance.

Figure 4 highlights the effect of retraining of the proposed methods compared to SCRUB across multiple epochs, for all three datasets.

Retraining the unlearned models on the retain set does not affect their performance significantly. The performance of the baseline on the other hand increases after an initial drop in case of CIFAR-100 and LACUNA-100. The initial drop may be attributed to the damage to the retain set performance caused by the initial *max-steps*. The subsequent increase can be attributed to the fact that the SCRUB training objective also optimizes the task loss on the retain set. Thus, once the model unlearns the forget set, SCRUB shifts the model capacity towards better learning the retain set. For CIFAR-10 this results in the model performing better than the DKVB models on the retain set as the retain set test accuracy after unlearning is higher than the original model. However, the baseline is unable to recover its original performance for CIFAR-100 and LACUNA-100. We refer to Appendix B for a similar comparison on the forget set.

## 6 LIMITATIONS AND FUTURE WORK

The proposed methods inherit the limitations of the DKVB (Träuble et al., 2023). Two important limitations are 1.) the reliance of DKVB on pre-trained encoders which can extract meaningful shared representations and 2.) trade-offs in downstream performance due to the use of an information bottleneck. Extensions to the model may involve training sparse representations inducing discrete bottleneck end-to-end. Further, in our experiments, we consider the setting of *class incremental* learning where the forget set can be easily identified and isolated. However, this may not always be true for a given task and more complicated approaches might be needed to identify the data that needs to be removed from the model. Thus, future work may involve scaling the proposed methods to such tasks and developing methods to identify and isolate the forget set in such cases.

## 7 CONCLUSION

In this work, we proposed a new approach to unlearning that requires negligible additional computation in order to unlearn a subset of data. This approach is based on the use of a discrete architectural bottleneck which induces sparse representations. These sparse representations facilitate unlearning a subset of data from the model with minimal to no performance drop on the rest of the data. We focused on the setting of class unlearning and our experiments show that the proposed approach, while being compute efficient, performs competitively with or in some cases better than a state-of-the-art approach which requires additional compute to perform unlearning. We also studied a compute-matched condition in which we allowed additional retraining. We found that the proposed approach did not benefit from such retraining, indicative of the fact that knowledge is highly localized in the Key-Value Bottleneck. Consequently, excising the activated key-value pairs from the model is a highly effective means of unlearning the forget set without disrupting the retain set.

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

## A    DECIDING THE FORGET CLASS

We assume that this class should be the most difficult one for the model to forget. Figures 5(a) - 5(c) show the number of mis-classifications per class on the test data, for all three datasets. For CIFAR-10, class #1 is the best-learned class with the lowest number of mis-classifications. Thus, we select class #1 as the forget class for the dataset. Similarly, for CIFAR-100 class 58 is the best-learned class and for LACUNA-100, class 48 is one of the best-learned classes with zero mis-classifications. Hence, we select classes #58 and #48 as the forget classes for CIFAR-100 and LACUNA 100 respectively. We use the same forget classes for experiments on the models with a linear layer in place of the DKVB (i.e., the baseline) as well.

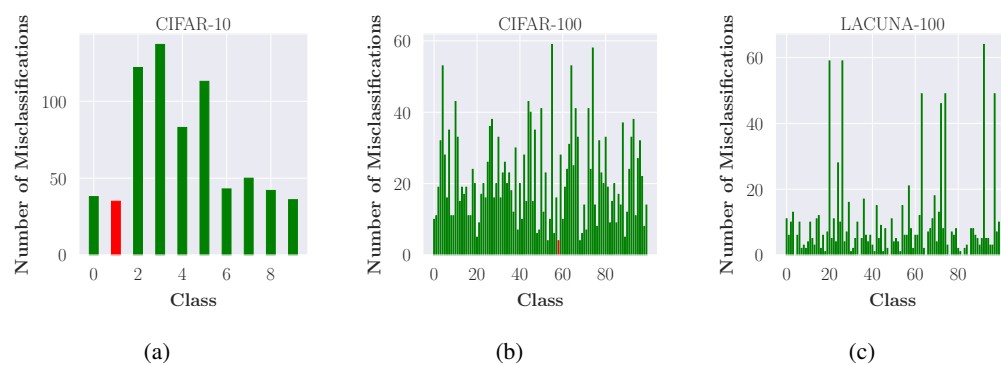

(a)                              (b)                              (c)

Figure 5: Number of mis-classifications per class for the test data. The red bars correspond to the class with the least number of mis-classifications (a) CIFAR-10: Class 1 has the least number of mis-classifications (b) CIFAR-100: Class 58 has the least number of mis-classifications (c) LACUNA-100: Classes 34, 48, 65, 76, 82 and 85 have 0 mis-classifications and hence, do not have a bar

## B    PERFORMANCE ON FORGET CLASS DURING RE-TRAINING

In all three cases, the baseline completely unlearns the forget set quickly.

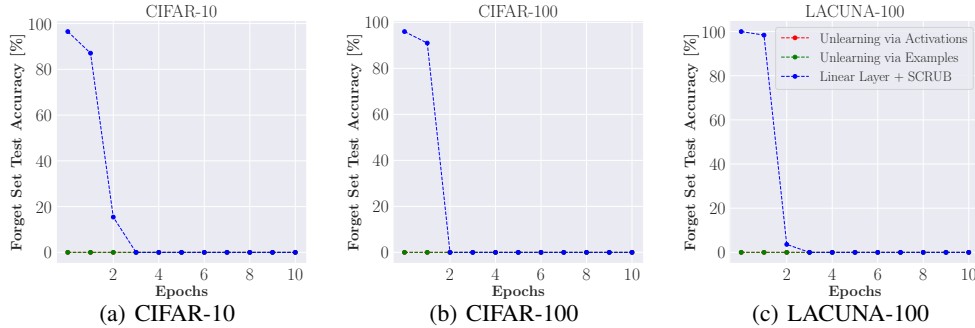

(a) CIFAR-10       (b) CIFAR-100       (c) LACUNA-100

Figure 6: Comparison between the performance of proposed methods with added compute and the baseline on the and forget set test data. Note that for the proposed methods, the plots start from after the initial zero shot unlearning. For the baseline, the plots start from the original models. The green line occludes the red line since both of them stay at 0% throughout the training.

## C  DETAILED ANALYSIS ZERO-SHOT UNLEARNING VIA ACTIVATIONS

### C.1  UNLEARNING VIA ACTIVATIONS

**CIFAR-10:** For CIFAR-10, we first validate the unlearning with $N_a \in \{10000n : n \in [0, 20]\}$ and record the performance on $\mathcal{D}_{test}^{retain}$ and $\mathcal{D}_{test}^{forget}$. Figure 2(a) shows the scatterplot of forget set test accuracy vs. retain set test accuracy for different values of $N_a$. We start off with $N_a = 0$, i.e., no unlearning, to $N_a = 300000$. The initial forget and retain set test accuracies for CIFAR-10 as shown in Table 1 are 93.02% and 96.50% respectively. As $N_a$ increases, the forget class test accuracy decreases, slowly at first and rapidly later. The model reaches random accuracy (i.e. 10% for CIFAR-10) on the forget class test data at $N_a = 150000$. At this point the retain set test accuracy is 92.97%. The model unlearns the forget class completely between $N_a = 170000$ (0.4%) and $N_a = 180000$ (0%). The retain set test accuracy at $N_a = 180000$ is 92.94%, which is almost similar to the initial accuracy of 92.61%. Further increasing $N_a$ till 200000 leads to an increase in retain set test accuracy to 93%.

**CIFAR-100** We vary $N_a$ for CIFAR-100 as $N_a \in \{5000n : n \in [0, 18]\}$. Figure 2(b) shows the scatterplot of retain set test accuracy vs. forget set test accuracy for these values. We start with retain set test accuracy of 78.24% and forget set test accuracy of 96% (see Table 1. On increasing $N_a$, the retain set test accuracy fluctuates in a small range around the original accuracy, reaching a maximum of 78.32% at $N_a = 40000$ and 450000. The forget set test accuracy drops slowly at first, dropping by 12% till $N_a = 20000$. The model reaches random test accuracy (1% for CIFAR-100) on the forget set at $N_a = 65000$ where the forget set test accuracy and retain set test accuracies are 1% and 78.14% respectively. It reaches 0% at $N_a = 70000$. At this point, the retain set test accuracy is 78.08%. Thus the model incurs a 0.2% deterioration in its retain set test accuracy by the time it has completely unlearned the forget class.

**LACUNA-100** We vary $N_a$ for LACUNA-100 as $N_a \in \{10000n : n \in [0, 20]\}$. Figure 2(c) shows the scatterplot of retain set test accuracy vs forget set test accuracy for LACUNA-100. The initial retain set test accuracy and forget set test accuracy for LACUNA-100 are 90.28% and 100% respectively (see Table 1. As we increase $N_a$, the retain set test accuracy increases monotonically to 90.43% till $N_a = 40000$ and then decreases before starting to fluctuate. Unlike CIFAR-10 and CIFAR-100, the decrease in forget set test accuracy is rapid from the beginning and slows down towards the end. The forget set test accuracy reaches a near random accuracy of 2% at $N_a = 140000$ where the retain set test accuracy is 90.24%. The model fully unlearns the forget set (i.e., forget set test accuracy = 0%) at $N_a = 150000$ at which point the accuracy of the model on the retain set test data is 90.13%. Thus there is only a 0.17% drop in performance as compared to the original model. Moreover as seen in Figure 2(c), the retain set test accuracy increases slightly as $N_a$ is increased further.

### C.2 UNLEARNING VIA EXAMPLES

**CIFAR-10** CIFAR-10 contains 5000 examples in the forget set training data. We unlearn randomly sampled $N_e$ examples out of these 5000 examples. We experiment with the following values of $N_e \in \{200n : n \in [0, 20]\}$. Figure 3(a) plots the forget set test performance vs. retain set test performance of the model for different values of $N_e$. We start off with retain set test and forget set test accuracies of 92.61% and 96.50% respectively. Similarly to in unlearning via activations, the forget class test accuracy decreases with an increase in $N_e$, and the decrease is slow at first and then rapid. The retain set test accuracy, on the other hand, increases monotonically, although marginally. The model achieves random accuracy on the forget class between $N_e = 2400$ and $N_e = 2600$, where the accuracies are 18.04% and 6.68% respectively. The accuracy on retain set test data at these points are 92.95% and 92.98% respectively. The forget set test accuracy drops to 0% (i.e., complete unlearning) between $N_e = 3000$ (0.08%) and $N_e = 3200$ (0%). The retain set test accuracies at these two points are 92.99% and 93.02%. Further increasing $N_e$ does not affect the retain set test accuracy much.

**CIFAR-100** CIFAR-100 contains 500 examples of the forget class in the training data. We vary $N_e$ for CIFAR-100 as $N_e \in \{10n : n \in [0, 15]\}$. Figure 3(b) shows the plot of forget set test accuracy vs. retain set test accuracy. The forget set test accuracy drops rapidly and monotonically from $N_e = 30$ to $N_e = 110$ where its value is 5% and the retain set test accuracy is 77.99%. It reaches the near random accuracy on 1.6% in the forget set test data at $N_e = 120$ where the retain set test accuracy is 78.01%. The model completely unlearns the forget set at $N_e = 150$ where the retain set test accuracy is 77.96%. The retain set test accuracy first decreases (almost monotonically) till $N_e = 90$. It increases thereafter slightly, before decreasing again towards the end.

**LACUNA-100** LACUNA-100 has 400 forget class training examples. We unlearn a randomly selected subset of size $N_e$ from it. $N_e$ is varied as $N_e \in \{20n : n \in [0, 15]\}$. Figure 3(c) shows the plot of retain set test accuracy to forget set test accuracy for different $N_e$. As in the case of unlearning via examples and unlike CIFAR-10 and CIFAR-100, the forget set test accuracy decreases rapidly with $N_e$ initially till $N_e = 60$; and rather slowly after that. The retain set test accuracy increases slightly at first and then starts fluctuating. The model reaches a random performance of 1% on the forget set test data at $N_e = 260$. The retain set test performance at this point is 90.25%. Finally, the model unlearns the forget class completely at $N_e = 300$, where the retain set test accuracy is 90.2%. Thus, the retain set performance drops by 0.09% by the time the model has completely unlearned the forget class.

## D USING THE PROPOSED METHODS AGAINST MEMBERSHIP INFERENCE ATTACKS

Depending on the application, complete unlearning of the forget set may not always be the final goal of unlearning. For several use cases such as removing information about corrupted data from the model or removing harmful biases exhibited by the model, maximal error on the forget set is desirable. However, for applications such as Differential Privacy, it is more desirable to achieve a forget set error which is similar to that of a model trained from scratch only on the retain set. Otherwise, it makes the unlearned model susceptible to Membership Inference Attacks (MIA) (Shokri et al., 2017). Although we do not explore this setting in this work, the proposed method can also be used for applications where complete unlearning is not desirable. This can be done by following a procedure similar to SCRUB+R (Kurmanji et al., 2023), wherein instead of selecting a particular model checkpoint, one can select the model corresponding to specific values of $N_a$ or $N_e$ such that the error on the forget set test data is similar to the reference point as defined in Kurmanji et al. (2023).

First, it is important to clarify that the proposed approach is not a-priori suited for selective unlearning, i.e. the setting where we want the model to forget specific examples or a small subset of examples instead of removing the information about an entire class. The KV bottleneck induces clusters of representation, where the members of a particular cluster correspond to the representations belonging to the same class. When we try to unlearn the representations corresponding to one particular example belonging to a particular class, the KV bottleneck routes the selection to other (key-)representations within the same cluster. Since these representations also contain information

about the same class as the examples we intend to unlearn, the model would still predict the class to be unlearnt.

Due to the same reason our approach is also not designed for working against traditional Membership Inference attacks. According to the basic attacks setup as explained in Kurmanji et al. (2023), the objective is to obtain a model that has unlearnt a small subset of specific examples (i.e. selective unlearning) such that the loss of the model on the unlearnt subset of examples should be indistinguishable from loss on examples that the model never saw during training.

Nevertheless, we attempt to modify the above setup to that of "Class Membership Inference Attacks (CMIA)". In CMIA, the aim is to defend against an attacker whose aim is to determine whether a model that has undergone unlearning ever saw a particular class as a part of its training data. Thus, we want the model to unlearn a particular class such that the losses/performance of the model on the unlearnt class is indistinguishable from a held-out class that the model never saw during its training. We describe the experimental setup and results below.

**Experimental Setup** We perform the experiment for CIFAR10. We divide the dataset into training data ($D_{Train}$), validation data ($D_{Val}$) and test data ($D_{Test}$). Training Data consists of 4000 examples per class; validation and test data consist of 1000 examples per class. We first trained a model on the first 9 classes of CIFAR10. Thus, class number 10 is the held-out class. Next, we unlearn class 1 from the model using the Unlearning via Activations approach introduced in the paper. We unlearn the model until the loss of the model on the validation sets of the forget class and the held-out class are similar. In our experiments, we find that we reach this point at approximately $N_a = 240000$. The loss $l(x, y)$ in our case would be the cross-entropy loss.

Next, we label the losses corresponding to the validation and test set of the forget class as 1 and those corresponding to the validation and test set of the held-out class as 0. We train a binary classifier on the validation losses of the forget and held-out sets and evaluate it on the test losses. We follow a similar setting for the baseline model, where we obtain the model suitable for MIA defense by using SCRUB+R (Kurmanji et al., 2023). For a successful defense, we would want the accuracy of the classifier to be close to 50%, indicating that it is unable to distinguish between the unlearnt class and the held-out class. Same as Kurmanji et al. (2023), we use `sklearn.logistic_regression` as our attacker (the binary classifier). We call the approach described above Partial UvA (Partial Unlearning via Activations). We run experiments for 3 random seeds, and the mean of the attacker performance is reported. Note that a similar procedure can also be followed using Unlearning via Examples.

**Observations and Results**: We report the results of the experiment described above in the table given below. We observe that although the baseline performs better, the proposed approach performs competitively, even though we have not intended to develop the method for this scenario.

Table 3: Comparison on Class Membership Inference Attacks between the proposed approach and the baseline. A binary classifier is trained on the validation losses of the forget and held-out sets and is evaluated on the test losses. The proposed approach performs competitively to SCRUB + R/

| Approach | Attacker Accuracy |
|---|---|
| Partial UvA | 53.50% |
| Linear Layer + SCRUB + R | **51.50%** |

# E  TRAINING DETAILS AND HYPERPARAMETERS

We do not use any data augmentation in any experiment. The transforms used are the same as CLIP (Radford et al., 2021) pretrained ViT/B-32 transforms.

## E.1  TRAINING DETAILS AND HYPERPARAMETERS FOR TRAINING THE ORIGINAL DKVB MODELS

Table 4 shows the hyperparameters used for training the base DKVB models

Table 4: Hyperparameters used for training the base DKVB models

|  | CIFAR-10 | CIFAR-100 | LACUNA-100 |
| --- | --- | --- | --- |
| top-k | 1 | 10 | 10 |
| Key Dimension | 8 | 8 | 8 |
| # of Key Init Epochs | 10 | 10 | 10 |
| Type of Value Init | Gaussian Random | Zeros | Uniform Random |
| # of Codebooks | 256 | 256 | 256 |
| # of Key-Value Pairs per Codebook | 4096 | 4096 | 4096 |
| Optimizer | Adam | Adam | Adam |
| LR | 0.1 | 0.3 | 0.3 |
| Batch Size | 256 | 256 | 256 |
| Epochs | 74 | 71 | 7 |

## E.2 TRAINING DETAILS AND HYPERPARAMETERS FOR TRAINING THE ORIGINAL BASELINE MODELS

For the baseline models, we deliberately train them to similar test ($\mathcal{D}_{test}$) accuracies as the models with a Discrete Key Value Bottleneck to ensure a fair comparison for unlearning. Table 5 shows the hyperparameters used for training the baseline models

Table 5: Hyperparameters used for training the baseline models

|  | CIFAR-10 | CIFAR-100 | LACUNA-100 |
| --- | --- | --- | --- |
| LR | 0.001 | 0.01 | 0.01 |
| Batch Size | 256 | 256 | 256 |
| Epochs | 2 | 2 | 2 |

## E.3 TRAINING DETAILS AND HYPERPARAMETERS FOR SCRUB

For the baseline, we run SCRUB on the model with linear layer. One *epoch* consists of one *min step* and may or may not contain a *max step*. One *max step* is included in every epoch for the first *msteps* epochs. We tune the hyperparameter *msteps* in our experiments and pick the case where the model is able to best recover its performance on the retain set test data and consider this model as the final unlearned model.We mention the hyperparameters used for running SCRUB for corresponding to the results presented in Section 5.3 in Table 6. In this case, training of SCRUB is stopped when either (a) the forget set accuracy has dropped to 0% without damaging the retain set accuracy or (b) at the end of 10 epochs. In the latter case, the value of the retain set test accuracy reported is the highest value in the training when under the constraint that the forget set test accuracy is 0 at that point. Results presented in Section 5.4 also uses the same set of hyperparameters except *min-step* which is always 10 since we train all the methods for 10 epochs. We do not consider cases where SCRUB is not able to completely unlearn the forget class within the given computational budget.

## E.4 TRAINING DETAILS AND HYPERPARAMETERS FOR RETRAINING EXPERIMENTS

Once the DKVB models are unlearned using *Unlearning via Activations* and *Unlearning via Examples*, we retraining them in order to make a fair comparison with the baseline. Thus, during retraining, the initial performance of these models on the retain set is same as the final performance of the zero-shot unlearned models. Table 7 show the hyperparameters used for retraining the unlearned DKVB models.

Table 6: Hyperparameters for SCRUB + Linear Layer Experiments shown in Section 5.3

|  | CIFAR-10 | CIFAR-100 | LACUNA-100 |
|---|---|---|---|
| Forget Set Batch Size | 256 | 256 | 256 |
| Retain Set Batch Size | 256 | 256 | 256 |
| # of max-steps (*msteps*) | 3 | 9 | 5 |
| # of min-steps | 3 | 10 | 7 |
| LR | 0.001 | 0.01 | 0.01 |
| Optimizer | Adam | Adam | Adam |
| Batch Size | 256 | 256 | 256 |

Table 7: Hyperparameters used for re-training experiments. **UvA** stands for *Unlearning via Activations* and **UvE** stands for *Unlearning via Examples*

|  | CIFAR-10 | | CIFAR-100 | | LACUNA-100 | |
|---|---|---|---|---|---|---|
|  | UvA | UvE | UvA | UvE | UvA | UvE |
| LR | 0.3 | 0.3 | 0.1 | 0.1 | 0.1 | 0.3 |
| Optimizer | Adam | Adam | Adam | Adam | Adam | Adam |
| Batch Size | 256 | 256 | 256 | 256 | 256 | 256 |
| Gradient Clipping | 0.1 | 0.1 | 0.1 | 0.1 | 0.1 | 0.1 |

## F COMPARISON OF RUNTIMES

In this section, we present a comparison of runtimes of our approach against the baseline, i.e. SCRUB (Kurmanji et al., 2023) as a proxy for comparing the compute requirements of the two approaches. Table 8 compares the runtimes of the proposed approaches against the baseline. We observe that the runtime for the proposed approaches is orders smaller than that of the baseline.

Table 8: Comparison of runtimes between the proposed methods and the baseline on CIFAR-10, CIFAR-100 and LACUNA-100.

|  | CIFAR-10 | CIFAR-100 | LACUNA-100 |
|---|---|---|---|
| Runtimes (in seconds) |  |  |  |
| DKVB via Activations (sec 5.2) | **5.02** | **1.57** | **2.74** |
| DKVB via Examples (sec 5.2) | 13.98 | 6.65 | 13.03 |
| Linear Layer + SCRUB | 288.80 | 921.56 | 553.31 |

## G ADDITIONAL RELATED WORKS

Jia et al. (2023) and Mehta et al. (2022) investigate machine unlearning in context of model sparsity in context of sparsity. Jia et al. (2023) leverages the Lottery Ticket Hypothesis Frankle & Carbin (2018) by using parameter pruning on a trained dense model to identify the token subnetwork. They observe that applying standard unlearning approaches to a sparsified networks is better as compared to doing unlearning directly on the dense network. Mehta et al. (2022) identify the Markovian Blanket of parameters corresponding to the examples to be unlearnt and updates those parameters. Thus, their approach can be seen as applying sparse updates to the network for unlearning. Both these approaches start with dense trained models and leverage sparsity for unlearning. Whereas the approaches proposed in this paper suggest using sparsity as an inductive bias in the model during the initial training along which combined with the architechtural prior of the DKVB makes it suitable for unlearning involving minimal compute requirements. Jia et al. (2023) on the hand spasifies the model after it has been trained. Mehta et al. (2022) involves sparse updates to the model parameters

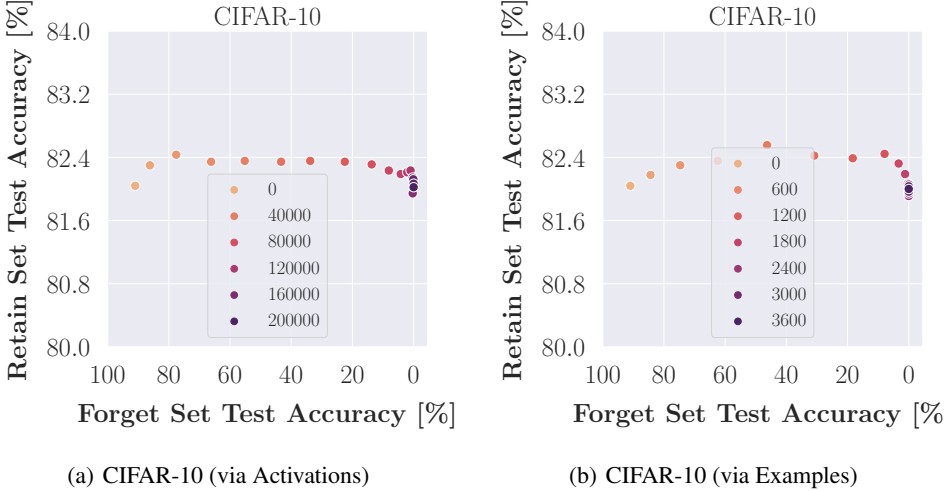

(a) CIFAR-10 (via Activations)   (b) CIFAR-10 (via Examples)

Figure 7: Performance on the retain set test data vs. Performance on the forget set test data for DKVB models with a supervised pretrained ResNet50 backbone on CIFAR-10.

as discussed previously. However, these sparse updates are utilized during unlearning as opposed to during training of the original model in the proposed approaches.

Warnecke et al. (2021) focus on unlearning in deleting information at the features level rather than unlearning specific instances or classes. They do so by using an optimization objective that incentivizes a model to replace the information about a set of features with a perturbed set of features. The proposed approaches on the other hand focus on unlearning entire classes of data. Chen et al. (2023), similarly to us, focuses on class unlearning. Unlearning is done by destroying the decision boundary of the forget class. The authors propose two boundary shift methods termed as *Boundary Shrink* and *Boundary Expanding*.

## H   EXPERIMENTS WITH A RESNET BACKBONE

To demonstrate the the proposed approaches are agnostic to the choice of the backbone, we run the same set of experiments presented in Section 5.2 and Section 5.3 on CIFAR-10 with an ImageNet supervised pretrained backbone. Figure 7 shows the scatter plot of retain set test performance vs. forget set test performance on CIFAR-10 with a ResNet backbone. Table 9 shows the comparison against baseline.

Table 9: Comparison between the proposed methods with a pretrained ResNet50 backbone and the baseline on CIFAR-10. We compare the change in performance on the retain and forget test data relative to the originally trained models. For the baseline, we report two cases: Case B where the model unlearns the forget set completely but the retain set performance is not preserved very well, and Case A where the retain set performance is not preserved very well, but the model does not unlearn the forget set completely

|  | CIFAR-10 | |
| --- | --- | --- |
| Rel. change from base model | $\mathcal{D}_{test}^{retain}$ | $\mathcal{D}_{test}^{forget}$ |
| DKVB via Activations (sec 5.2) | 0.04% | **-100%** |
| DKVB via Examples (sec 5.2) | -0.07% | **-100%** |
| Linear Layer + SCRUB (A) | **0.44%** | -96.04% |
| Linear Layer + SCRUB (B) | -11.75% | **-100%** |

There is a steep but drop in the retain set test accuracy towards the end when $N_a$ and $N_e$ are high enough. However, speaking in absolute terms, the drop seems to be not significant.

# I EXPERIMENTAL RESULTS ON IMAGENET-1K

Table 10: Comparison between the proposed methods and the baselines on ImageNet-1k. We compare the change in performance on the retain and forget test data relative to the originally trained models.

| Rel. change from base model | $\mathcal{D}_{test}^{retain}$ | $\mathcal{D}_{test}^{forget}$ |
|---|---|---|
| DKVB via Activations (sec 5.2) | 0.15% | -100% |
| DKVB via Examples (sec 5.2) | -0.03% | -100% |
| Linear Layer + SCRUB | **7.31%** | -100% |

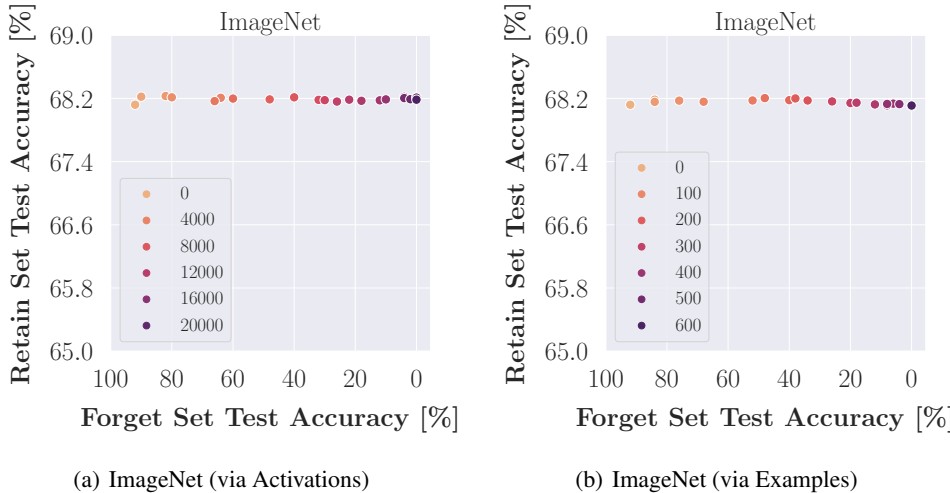

(a) ImageNet (via Activations)          (b) ImageNet (via Examples)

Figure 8: Performance on the retain set test data vs. Performance on the forget set test data for DKVB models on ImageNet-1k

