# OpenReview forum: "Unlearning via Sparse Representations"
_ICLR.cc/2024/Conference — Submitted to ICLR 2024_

### Official Review · Reviewer_ffna · 2023-10-31

**Soundness:** 3 good
**Presentation:** 3 good
**Contribution:** 2 fair
**Rating:** 5
**Confidence:** 3

**Summary:**

In this paper, the authors propose a compute-free method for achieving class unlearning in deep models. Their method assumes that the model is trained with a Discrete Key-Value Bottleneck (DKVB) which maps input data to key-value pairs. By removing the key corresponding to a certain class, the model is no longer able to provide a correct classification for it.

**Strengths:**

- The method has virtually no computation overhead (just a few inference steps)

- It does not introduce any new parameters

- It provides an effective way for unlearning an entire class from a pretrained classification model

**Weaknesses:**

- Following Xu et al., 2023, the method falls in the "Weak Unlearning" category; meaning that the unlearning only affects the final activations of the models (in this case we can identify it with the DKVB).

- Connected to the point above, weak unlearning does not guarantee that the internal parameters of the model do not still encode information about the forget set after the unlearning. In the context of this paper, given that the backbone of the model is always kept frozen and no finetuning is necessary, one could argue that, if the pretraining is done on the same dataset (Tab E.1), the final model will still contain forget samples encoded in its weights. This of course is a matter of current research, however it also limits the applicability of the proposed method.

- I wonder if this method would be similar to taking a pretrained model (without DKVB) and then just either *i)* retraining the classification layer without the forget class or *ii)* masking out the forget class from the output logits. In both cases, I would find it difficult to call it "unlearning" but rather "obfuscation" of some kind, although I am not an expert in this field.

- Unlearning an entire class seems perhaps unrealistic in practice; whereas unlearning specific instances should be more relevant.

- I think that the experimental evaluation is a bit lacking and could be expanded to larger datasets such as ImageNet, as the proposed method is almost compute-free

**Questions:**

- How does your method compare to SCRUB in terms of the categorization provided by Xu et al. ?

- What information about the forget samples remain after removing the keys?

- Could you provide some practical examples in which: 1) whole class unlearning is relevant 2) weak unlearning is acceptable under law requirements (e.g. GDPR)

- Can your method unlearn just a subset of the forget class? e.g. perhaps by using denser keys in the DKVB? If so, some experiments presenting this result would greatly improve the impact of this work.

- Would your method work also if the pretrain dataset is different from the target one?

- Can you expand more on Sec.D of the Appendix, as I think it is relevant from a practical point of view?

---

> ### Author Response · Authors · 2023-11-22
> **Response to Reviewer ffna**
>
> We thank the reviewer for taking the time go through our work and giving a detailed feedback as well as suggestions. We hope to answer the reviewers questions in the rebuttal that follows:
>
> **Regarding the evaluation of the approach at scale**
>
> We evaluate our approach on a larger scale by training the baseline and DKVB models on the ImageNet-1k and attempting to unlearn class number 1.
>
> * Experimental Setup - We train the DKVB and baseline model (i.e. Frozen backbone + trainable linear layer) on ImageNet-1k up to 68% classification accuracy. ImageNet-1k consists of 1.28 million training examples and 50000 test examples divided into 100 classes. Next, we choose class 1 as the class to be unlearnt, and perform the same set of experiments as performed for the three datasets in Section 5.2 and Section 5.3 in the paper.
>
> * Results - We present a comparative analysis between the baseline and the proposed approach on ImageNet-1k in the table given below. Further, we have also included scatter plots for the retain set accuracy vs forget set accuracy in Appendix I of the paper. We observe that our approach performs equally well on ImageNet as compared to the three datasets already included in the paper, with similar hyperparameters This demonstrates that the proposed approach is equally effective in large-scale settings as well. The given table presents a comparison between the proposed methods and the baselines on ImageNet-1k. We compare the change in performance on the retain and forget test data relative to the originally trained models.
>
> |  | $D_{retain}^{test}$ | $D^{forget}_{test}$ |
> |----------|----------|----------|
> | DKVB via Activations (sec 5.2) | 0.15 % | $\textbf{-100}$ % |
> | DKVB via Examples (sec 5.2) | -0.03 % | $\textbf{-100}$ % |
> | Linear Layer + SCRUB | $\textbf{7.31\}$ % | $\textbf{-100}$ % |
>
> **Regarding Instance-Specific Unlearning**
>
> In this work, we focus on the setting of class unlearning. Instance-specific unlearning (or selective unlearning) would be non-trivial with the proposed approach and we leave investigations regarding that for future work.
>
> **Elaborating on Membership Inference Attacks (Appendix Section D)**
>
> First, we would like to clarify that our approach is not a-priori suited for selective unlearning, i.e. the setting where we want the model to forget specific examples or a small subset of examples instead of removing the information about an entire class. The KV bottleneck induces clusters of representation, where the members of a particular cluster correspond to the representations belonging to the same class. When we try to unlearn the representations corresponding to one particular example belonging to a particular class, the KV bottleneck routes the selection to other (key-)representations within the same cluster. Since these representations also contain information about the same class as the examples we intend to unlearn, the model would still predict the class to be unlearnt.
>
> Due to the same reason our approach is also not designed for working against traditional Membership Inference attacks. According to the basic attacks setup as explained in their paper  the objective is to obtain a model that has unlearnt a small subset of specific examples (i.e. selective unlearning) such that the loss of the model on the unlearnt subset of examples should be indistinguishable from examples that the model never saw during training.
>
> Nevertheless, we attempt to modify the above setup to that of "Class Membership Inference Attacks (CMIA)". In CMIA, the aim is to defend against an attacker whose aim is to determine whether a model that has undergone unlearning ever saw a particular class as a part of its training data. Thus, we want the model to unlearn a particular class such that the losses/performance of the model on the unlearnt class is indistinguishable from a held-out class that the model never saw during its training. We describe the experimental setup and results below:
>
> * Experimental Setup: We perform the experiment for CIFAR10. We divide the dataset into training data ($\mathcal{D_{Train}}$), validation data ($\mathcal{D_{Val}}$) and test data ($\mathcal{D_{Test}}$). Training Data consists of 4000 examples per class; validation and test data consist of 1000 examples per class. We first trained a model on the first 9 classes of CIFAR10. Here, class number 10 is the held-out class. Next, we unlearn class 1 from the model using the Unlearning via Activations approach introduced in the paper. We unlearn the model until the loss on the validation sets of the forget class and the held-out class are similar. In our experiments, we find that we reach this point at approximately $N_a = 240000$. The loss $l(x, y)$ in our case would be the cross-entropy loss.

---

> ### Author Response · Authors · 2023-11-22
> **Response to Reviewer ffna [Continued]**
>
> * Next, we label the losses corresponding to the validation and test set of the forget class as 1 and those corresponding to the validation and test set of the held out class as 0. We train a binary classifier as done in the "Basic Attack" setting in [6] on the validation losses of both sets and test it on the test losses. We follow a similar setting for the baseline model, where we obtain the model suitable for MIA defense by following a procedure similar to SCRUB+R as introduced in [6]. For a successful defense, we would want the accuracy of the classifier to be close to 50%, indicating that it is unable to distinguish between the unlearnt class and the held-out class. Same as Kurmanj [1], we use sklearn.logistic_regression as our attacker (the binary classifier). We call the approach described above Partial UvA (Partial Unlearning via Activations).  We run experiments for 3 random seeds, and the mean of the attacker performance is reported.
> * Observations and Results: We report the results of the experiment described above in the table given below. We observe that although the baseline performs better, the proposed approach performs competitively, even though we have not intended to develop the method for this scenario.
>
> | Approach | Accracy of the attacker |
> |-----------|-----------|
> | Partial UvA | 53.5 |
> | Linear Layer + SCRUB + R | 51.50 |
>
> **Would the method work if the pretrain dataset is different from the target one**
>
> In all of our experiments, the dataset used for pretraining the backbone is always distinct from the downstream task. For all the experiments in the paper, we use a ViT-B/32 backbone which has been pretained using CLIP (Radford et al., 2021). which uses web-scale Image-Text data for pretraining. In our new experiments with a ResNet50 backbone (see Appendix H) which has been pretrained in a supervised manner on ImageNet and the tasks used for downstream tasks are CIFAR-10, CIFAR-100 and LACUNA-100).

---

### Official Review · Reviewer_g9Bk · 2023-11-01

**Soundness:** 2 fair
**Presentation:** 2 fair
**Contribution:** 2 fair
**Rating:** 5
**Confidence:** 2

**Summary:**

Nonetheless, every approach to machine unlearning necessitates a substantial increase in computational resources to facilitate the unlearning process. In this scholarly work, the authors posit that neural information bottlenecks offer a highly efficient and precise method for unlearning. Their research demonstrates that this proposed technique effectively unlearns the specified data, causing minimal disruption to the model's performance on the remaining dataset. The researchers assess the effectiveness of their approach in the context of class unlearning, using three distinct datasets: CIFAR-10, CIFAR-100, and LACUNA100. They conduct a comparative analysis between their proposed technique and SCRUB, a state-of-the-art unlearning approach that employs knowledge distillation.

**Strengths:**

1. Zero Shot Unlearning.
2. Low computational cost
3. Model Specific only applicable to models with Discrete Key Value Bottleneck.

**Weaknesses:**

1. Even though the whole motivation of this approach is to implement a new approach with better computational efficiency no experiments on computational cost are done. This approach is not measured with current approaches of class unlearning in terms of computational efficiency.

**Questions:**

1. Is this approach model agnostic? I am not sure if this can applied to other models if the bottleneck is not present.

---

> ### Author Response · Authors · 2023-11-22
> **Response to Reviewer g9Bk**
>
> We thank the reviewer for taking the time to go through our work. We attempt to address the points raised by the reviewer in this rebuttal:
>
> **Regarding Experiments around the computational cost of the approaches discussed**
>
> We measure the runtimes of the experiments discussed in Section 5.3 and Table 2 as a proxy to the computational costs required by the approaches discussed. The results are presented in the table below. We note that both the proposed approaches are several times faster than the baseline
>
> | | CIFAR-10 | CIFAR-100 | LACUNA-100 |
> |----------|----------|----------|----------|
> | Runtimes (in seconds) | | | |
> | DKVB via Activations (sec 5.2) | $\textbf{5.02}$ | $\textbf{1.57}$ | $\textbf{2.74}$ |
> | DKVB via Examples (sec 5.2) | 13.98 | 6.65 | 13.03 |
> | Linear Layer + SCRUB | 288.80 | 921.56 | 553.31 |
>
> We find that Unlearning via Activations is consistently the most compute efficient approach while both Unlearning via Activations and Unlearning via Examples being at least ~20x faster than the baseline
>
> **Regarding the approach being model agnostic**
>
> The reviewer correctly notes that the proposed approaches are conditioned on the presence of a Discrete Key Value Bottleneck. However, it is agnostic to the choice of the frozen backbone and the decoder architectures. This can be supported by evidence from our new experiments with a ResNet50 backbone (see Appendix Section H) which show that the proposed approach also works with backbones other than a CLIP pretrained ViT-B/32.

---

### Official Review · Reviewer_Pr1b · 2023-11-02

**Soundness:** 2 fair
**Presentation:** 2 fair
**Contribution:** 2 fair
**Rating:** 6
**Confidence:** 2

**Summary:**

EDIT: After reading the rebuttal, I have decided to raise my score. The techniques used in this work are not necessarily very novel, but it does manage to unlearn classes in a much more computationally efficient zero-shot manner than previous works.

This paper studies the problem of unlearning where the task is to forget a set of concepts learned during training while retaining good performance on other concepts. The paper demonstrates that sparse representations can help decompose the concepts, making post-hoc unlearning amendable. Specifically, they train an architecture with a Discrete Key-Value Bottleneck which specifically induces sparse key-value pairs. Next, given training examples that correspond with a particular concept, they propose pruning away certain key-value pairs that light up the most for these examples. On image classification tasks CIFAR10/100 and and LACUNA-100, they show that this procedure can drop a particular class’s test accuracy to 0% while retaining the same performance on the remaining classes.

**Strengths:**

The paper demonstrates strong performance gains, and is computationally inexpensive. The method only requires forward passes since irrelevant key-value pairs are simply pruned unlike previous methods which utilize some form of negative gradients updates or knowledge distillation.

The paper is generally an easy read and figures are clear.

**Weaknesses:**

In terms of novelty, I think it’s important to point out that this paper is a direct application of Discrete Key-Value Bottleneck (Trauble 2022).
1. DKVB was proposed for class-incremental learning, and class unlearning is quite literally the inverse task.
2. The original work also shows improvements on CIFAR10/100, the same benchmarks in this paper.
3. DKVB improves class-incremental learning since each class can be learned by disjoint key-value pairs, thus the model updates for learning new classes can be localized to certain parameters. Thus, why DKVB would help with unlearning might be quite trivial – simply deleting the key-value pairs corresponding with a certain class.

There are also a couple improvements that could be made to improve the quality of the paper.

1. The paper focuses on performance evaluation, but there’s a lack of empirical analysis of their conceptual motivation. For example, how often are key-value pairs shared between classes? Is it close to 0? Can this be visualized?
2. A more diverse range of benchmarks is necessary to test the unlearning capabilities of DKVB. It’s unclear how successful the unlearning method is because the evaluation benchmarks are quite similar to each other (CIFAR10/100, LACUNA100). It might be interesting to evaluate the method on more diverse benchmarks such as benchmarks with outlier classes that are quite similar to each other, or training to distinguish between superclasses and unlearning a specific class in a superclass.

**Questions:**

1. “Zero-shot” is thrown around in the paper, but seems like the wrong terminology here and I am unsure what it is referring to exactly since their method and the baseline method SCRUB both require utilizing training examples. Maybe they mean something closer to “zero-order”.
2. The authors choose to unlearn a class until 0% accuracy. Is this correct, or should you be unlearning up until random chance (10%)?
3. Section 5.4 was a bit unclear.
- What do you mean by “implications of additional compute”? Are you retraining the DKVB models for longer?
- What do you mean by “retraining”? In Figure 4, you state “retraining…using the proposed methods” (It says “proposed models” but I think you meant to say “proposed methods”?) What does it mean to retrain using SCRUB. Or is the SCRUB’ed model then retrained to learn the forgotten class just by normal supervised training?
- It seems a bit strange that the performance would ever drop during retraining with SCRUB.

---

> ### Author Response · Authors · 2023-11-22
> **Response to Reviewer Pr1b**
>
> We thank the reviewer for the insightful comments and suggestions. We attempt address the concerns raised by the reviewer below:
>
> **Evaluating the approach on more benchmarks**
>
> We tested the proposed approach on ImageNet-1k dataset and the results are as follows:
>
> * Experimental Setup - We train the DKVB and baseline model (i.e. Frozen backbone + trainable linear layer) on ImageNet-1k up to 68% classification accuracy. ImageNet-1k consists of 1.28 million training examples and 50000 test examples divided into 1000 classes. Next, we choose class 1 as the class to be unlearnt, and perform the same set of experiments as performed for the three datasets in Section 5.2 and Section 5.3 in the paper.
> * Results - We present a comparative analysis between the baseline and the proposed approach on ImageNet-1k in the table given below. Further, we have also included scatter plots for the retain set accuracy vs forget set accuracy in Appendix I of the paper. We observe that our approach performs equally well on ImageNet as compared to the three datasets already included in the paper, with similar hyperparameters This demonstrates that the proposed approach is equally effective in large-scale settings as well. The table given below provides a comparison between the proposed methods and the baselines on ImageNet-1k. We compare the change in performance on the retain and forget test data relative to the originally trained models.
>
> |  | $D_{retain}^{test}$ | $D^{forget}_{test}$ |
> |----------|----------|----------|
> | DKVB via Activations (sec 5.2) | 0.15 % | $\textbf{-100}$ % |
> | DKVB via Examples (sec 5.2) | -0.03 % | $\textbf{-100}$ % |
> | Linear Layer + SCRUB | $\textbf{7.31\}$ % | $\textbf{-100}$ % |
>
> **Complete Unlearning vs Unlearning until random accuracy**
>
> As discussed in the literature, unlearning completely vs random accuracy depends on the task at hand. For instance, in cases where we care about removing harmful biases from a model (e.g. remove a users data completely), complete unlearning is more desirable. In other scenarios such as Membership Inference Attacks (MIAs), the focus is on user privacy, it is desired that the model unlearns only to an extent such that the performance of the model on the unlearnt class is equivalent to that on a class that it has never seen. For the proposed approaches of Unlearning via Examples and Unlearning via Activations, this “extent of unlearning” can be controlled using the parameters $N_e$ and $N_a$ respectively as shown in Figures 2 and 3. Further, our new experiments of MIAs as discussed in Appendix D. Here, we show that the proposed approach is also useful in cases where complete unlearning is not desired.
>
> **Clarifications regarding Section 5.4**
>
> * Implications of using additional compute - The proposed approaches do not require additional compute for unlearning a class apart from the compute required for inference. The baseline, on the other hand, requires additional compute in the form of training the model and optimizing it w.r.t. a given objective. Thus, we attempt to investigate what would happen if the proposed approach also somehow made use of additional compute. In order to make use of some additional compute, once the DKVB models have undergone unlearning of the forget class, we train them on just the retain set in order to cover up any damage in performance on the retain set that might have occurred due to the unlearning.
> * Use of the term “retraining” - We use the term “retraining” to refer to the above-explained process of training the unlearnt DKVB models on only the retain set in order to improve the performance on the retain set.

---

### Official Review · Reviewer_GrC6 · 2023-11-04

**Soundness:** 2 fair
**Presentation:** 2 fair
**Contribution:** 2 fair
**Rating:** 5
**Confidence:** 4

**Summary:**

This paper introduces a novel approach to machine unlearning, aiming to efficiently remove data from a trained model—a process known as "forgetting"—without the significant computational overhead typically associated with such tasks. The authors propose a zero-shot unlearning technique that utilizes a discrete representational bottleneck to erase specific knowledge of a forget set from a model. This technique is claimed to preserve the overall performance of the model on the remaining data.

**Strengths:**

**Writing**:
The paper's logic is clear, and it is easy to follow.

**Weaknesses:**

- **Incremental Novelty**: While the application of concepts from DKVB [1] to the context of machine unlearning is interesting, it may appear as an incremental advance rather than a groundbreaking innovation. Nonetheless, the practical integration of these ideas to address real-world challenges is acknowledged as valuable and can sometimes be sufficient to make significant contributions to the field. (Minor points)

- **Comparative Analysis**: The paper could benefit from a broader comparison with existing work, particularly the zero-shot unlearning approach presented in [2]. Additionally, for the specific examples of unlearning demonstrated in Figure 3, it would be advantageous to include a wider range of baselines, as referenced in [3-6], to provide a more comprehensive evaluation of the proposed method.

- **Unlearning Scenarios**: The focus on 'complete unlearning' may limit the paper's scope, given the broader spectrum of unlearning scenarios presented in [2-7], where the goal is for the unlearned model to closely resemble a model that has been retrained without the forgotten data. Although the authors claim in Appendix D that their method is applicable to traditional machine unlearning problems, detailed results and discussions within these contexts would strengthen the paper's contributions.

- **Efficiency Metrics**: For claims regarding efficiency, the absence of reported running times is a noticeable omission. Including such metrics would substantiate the claims of the method's efficiency and offer a tangible comparison with other techniques.

- **Model Diversity**: The exclusive use of the ViT model in the experiments may raise questions about the method's generalizability. As DKVB [1] also considers ResNet architectures, a discussion on the application or limitation of the proposed method to other architectures would provide deeper insights into its adaptability and utility across various models.

- **Limited Literature Review**: Given the recent surge in research surrounding machine unlearning, the paper's literature review could be perceived as insufficiently comprehensive. I included some papers published recently in the following references.

>[1] Träuble, Frederik, et al. "Discrete key-value bottleneck." International Conference on Machine Learning. PMLR, 2023.
>
>[2] Chundawat, Vikram S., et al. "Zero-shot machine unlearning." IEEE Transactions on Information Forensics and Security (2023).
>
>[3] Chen, Min, et al. "Boundary Unlearning: Rapid Forgetting of Deep Networks via Shifting the Decision Boundary." Proceedings of the IEEE/CVF Conference on Computer Vision and Pattern Recognition. 2023.
>
>[4] Warnecke, Alexander, et al. "Machine unlearning of features and labels." arXiv preprint arXiv:2108.11577 (2021).
>
>[5] Jia, Jinghan, et al. "Model sparsification can simplify machine unlearning." arXiv preprint arXiv:2304.04934 (2023).
>
>[6] Kurmanji, Meghdad, Peter Triantafillou, and Eleni Triantafillou. "Towards Unbounded Machine Unlearning." arXiv preprint arXiv:2302.09880 (2023).
>
>[7] Golatkar, Aditya, Alessandro Achille, and Stefano Soatto. "Eternal sunshine of the spotless net: Selective forgetting in deep networks." Proceedings of the IEEE/CVF Conference on Computer Vision and Pattern Recognition. 2020.

**Questions:**

- Could you consider including the additional baselines identified as pertinent in Figure 3? This would help provide a comprehensive evaluation against a variety of existing methods.
- Would it be possible to provide more experimental results assessed against the conventional machine unlearning benchmarks, particularly those evaluating the similarity to a model retrained without the forget set?
- Can a comparison be drawn against the baseline referenced regarding zero-shot unlearning settings, as outlined in the weaknesses?
- Could you augment your experimental results with additional insights on how your technique performs on models within the ResNet family?
- Considering the broader discussion in the literature on the role of sparsity in machine unlearning, could you discuss how your approach aligns with or diverges from these concepts, specifically referring to studies [1-2]?

> [1] Mehta, Ronak, et al. "Deep unlearning via randomized conditionally independent hessians." Proceedings of the IEEE/CVF Conference on Computer Vision and Pattern Recognition. 2022.
>
> [2] Jia, Jinghan, et al. "Model sparsification can simplify machine unlearning." also mentions sparsity can help machine unlearning.

---

> ### Author Response · Authors · 2023-11-22
> **Response to Reviewer Reviewer GrC6**
>
> We thank the reviewer for their detailed and very helpful feedback on our paper. We hope  to address your questions and suggestions in the following.
>
>
> **Regarding Model Diversity**
>
> As requested by the reviewer, we performed the same set of experiments presented in the paper in Sections 5.2 and 5.3 for CIFAR-10, with a backbone from the ResNet family.
> * We use an ImageNet supervised pretained ResNet50  model as the backbone for our approach as well as the baseline model.
> * The results are as presented in the table below. We have also attached the scatter plots for the retain set accuracy vs forget set accuracy in Appendix H of the paper. We observe that the results with ResNet backbone are fairly similar to the ones with ViT backbone, discounting the difference in absolute performances. Importantly, even in this scaled-up setting we are able to perform unlearning with minimal compute without damaging the performance on the retain set.
> * We compare the change in performance on the retain and forget test data
> relative to the originally trained models. For the baseline, we report two cases: Case B where the
> model unlearns the forget set completely but the retain set performance is not preserved very well,
> and Case A where the retain set performance is not preserved very well, but the model does not
> unlearn the forget set completely
>
>
> |  | $D^{retain}_{test}$ | $D^{forget}_{test}$ |
> |----------|----------|----------|
> | DKVB via Activations (sec 5.2) | 0.04% | $\textbf{-100}$% |
> | DKVB via Examples (sec 5.2) | -0.07% | $\textbf{-100}$% |
> | Linear Layer + SCRUB (A) | $\textbf{-0.44\}$% | -96.04% |
> | Linear Layer + SCRUB (B) | -11.75% | $\textbf{-100}$% |
>
> **Regarding comparison of runtimes**
>
> We measured the runtimes of the experiments discussed in Section 5.3 and Table 2. The results are presented in the table below. We note that both the proposed approaches are several times faster than the baseline
>
> | | CIFAR-10 | CIFAR-100 | LACUNA-100 |
> |----------|----------|----------|----------|
> | Runtimes (in seconds) | | | |
> | DKVB via Activations (sec 5.2) | $\textbf{5.02}$ | $\textbf{1.57}$ | $\textbf{2.74}$ |
> | DKVB via Examples (sec 5.2) | 13.98 | 6.65 | 13.03 |
> | Linear Layer + SCRUB | 288.80 | 921.56 | 553.31 |
>
> **Regarding more Unlearning Scenarios**
>
> First, we would like to clarify that our approach is not a-priori suited for selective unlearning, i.e. the setting where we want the model to forget specific examples or a small subset of examples instead of removing the information about an entire class. The KV bottleneck induces clusters of representation, where the members of a particular cluster correspond to the representations belonging to the same class. When we try to unlearn the representations corresponding to one particular example belonging to a particular class, the KV bottleneck routes the selection to other (key-)representations within the same cluster. Since these representations also contain information about the same class as the examples we intend to unlearn, the model would still predict the class to be unlearnt.
>
> Due to the same reason our approach is also not designed for working against traditional Membership Inference attacks. According to the basic attacks setup as explained in Kurmanji et al., 2023, the objective is to obtain a model that has unlearnt a small subset of specific examples (i.e. selective unlearning) such that the loss of the model on the unlearnt subset of examples should be indistinguishable from examples that the model never saw during training.
>
> Nevertheless, we attempt to modify the above setup to that of "Class Membership Inference Attacks (CMIA)". In CMIA, the aim is to defend against an attacker whose aim is to determine whether a model that has undergone unlearning ever saw a particular class as a part of its training data. Thus, we want the model to unlearn a particular class such that the losses/performance of the model on the unlearnt class is indistinguishable from a held-out class that the model never saw during its training. We describe the experimental setup and results below:
>
> * Experimental Setup: We perform the experiment for CIFAR10. We divide the dataset into training data ($\mathcal{D_{Train}}$), validation data ($\mathcal{D_{Val}}$) and test data ($\mathcal{D_{Test}}$). Training Data consists of 4000 examples per class; validation and test data consist of 1000 examples per class. We first trained a model on the first 9 classes of CIFAR10. Here, class number 10 is the held-out class. Next, we unlearn class 1 from the model using the Unlearning via Activations approach introduced in the paper. We unlearn the model until the loss on the validation sets of the forget class and the held-out class are similar. In our experiments, we find that we reach this point at approximately $N_a = 240000$. The loss $l(x, y)$ in our case would be the cross-entropy loss.

---

> ### Author Response · Authors · 2023-11-22
> **Response to Reviewer GrC6 [Continued]**
>
> * Next, we label the losses corresponding to the validation and test set of the forget class as 1 and those corresponding to the validation and test set of the held out class as 0. We train a binary classifier as done in the "Basic Attack" setting in [6] on the validation losses of both sets and test it on the test losses. We follow a similar setting for the baseline model, where we obtain the model suitable for MIA defense by following a procedure similar to SCRUB+R as introduced in [6]. For a successful defense, we would want the accuracy of the classifier to be close to 50%, indicating that it is unable to distinguish between the unlearnt class and the held-out class. Same as Kurmanj [1], we use sklearn.logistic_regression as our attacker (the binary classifier). We call the approach described above Partial UvA (Partial Unlearning via Activations).  We run experiments for 3 random seeds, and the mean of the attacker performance is reported.
> * Observations and Results: We report the results of the experiment described above in the table given below. We observe that although the baseline performs better, the proposed approach performs competitively, even though we have not intended to develop the method for this scenario.
>
> | Approach | Accracy of the attacker |
> |-----------|-----------|
> | Partial UvA | 53.5 |
> | Linear Layer + SCRUB + R | 51.50 |
>
> **Regarding Limited Literature Review**
>
> We thank the reviewer for pointing us to these important papers. We have discussed these ([1], [3], [4], [5]) works as well as drawn a comparison of the proposed approach to [1] and [5] in the "Additional Related Works" section in the Appendix (section G).
>
>  > [1] Jia, Jinghan, et al. "Model sparsification can simplify machine unlearning." arXiv preprint arXiv:2304.04934 (2023). \
> > [2] Chundawat, Vikram S., et al. "Zero-shot machine unlearning." IEEE Transactions on Information Forensics and Security (2023). \
> > [3] Warnecke, Alexander, et al. "Machine unlearning of features and labels." arXiv preprint arXiv:2108.11577 (2021). \
> > [4] Chen, Min, et al. "Boundary Unlearning: Rapid Forgetting of Deep Networks via Shifting the Decision Boundary." Proceedings of the IEEE/CVF Conference on Computer Vision and Pattern Recognition. 2023.\
> > [5] Mehta, Ronak, et al. "Deep unlearning via randomized conditionally independent hessians." Proceedings of the IEEE/CVF Conference on Computer Vision and Pattern Recognition. 2022 \
> > [6] Kurmanji, Meghdad, Peter Triantafillou, and Eleni Triantafillor> "Towards Unbounded Machine Unlearning." arXiv preprint arXiv:2302.09880 (2023).

---

### Author Response · Authors · 2023-11-23
**General Comment**

We thank all the reviewers for going through our work and giving useful feedback and suggestions. We would like to summarize the new experiments that we performed and the corresponding additions to the paper.

**Experiments on Model Diversity**

* Requested by reviewer GrC6
* We perform same set of experiments as described in Sections 5.2 and 5.3 on CIFAR-10, but with a ImageNet supervised pretrained ResNet50.
* We observe that the proposed approach also works with a non CLIP-ViTB/32 backbone is is agnostic to the choice of the pretrained frozen backbone
* Added to Appendix Section H.

**Experiments on a Larger Scale (ImageNet-1k)**

* Requested by Reviewers Pr1b and ffna
* We run the experiments discussed in Sections 5.2 and 5.3 on the ImageNet-1k dataset
* We observe that the proposed approaches work equally well on large scale datasets such as ImageNet-1k
* Added to Appendix I

**Experiments for comparing compute cost with the baseline**

* Requested by reviewers GrC6 and g9Bk
* We measure the runtimes (as a proxy for the computational cost) for all experiments presented in Table 2.
* We observe that the proposed approach is at least 20x faster than the baseline
* Added to Appendix F

**Experiments for Membership Inference Attacks**

* Requested by reviewers GrC6 and ffna
* We run a variant of MIA experiments, called Class Membership Inference Attacks which is suitable for our approach. In CMIAs, the attack tries to determine where a given class of data was ever seen during the training of the original model and has been subsequently unlearnt.
* We observe that the proposed approach performs competitively on CMIA experiments.
* Added to Appendix D

Apart from these additional experiments, we have also attempted to answer each reviewer's other concerns and questions

---

### Meta-Review · Area_Chair_Skj4 · 2023-12-05

**Metareview:**

This paper presents a machine unlearning method based on the Discrete Key-Value Bottleneck (DKVB) approach.

The reviewers appreciate the paper for the simplicity and ease of implementation of the method, and the writing to be clear. However, there were several concerns as well. In particular, (1) the method is a simple application of DKVB and applying DKVB to the unlearning problem is somewhat straightforward which undermines the novelty aspect, (2) the notion of forgetting is that of "weak forgetting", and (3) missing results on computational efficiency.

While the authors' rebuttal tried to address some of the aspects (e.g., for computational efficiency, additional results were provided), it did not address concerns about novelty.

In the end, none of the reviewers voted for acceptance. From my own reading of the paper, I also concur with their assessment.

**Justification For Why Not Higher Score:**

Limited novelty and mostly straightforward application of DKVB.

**Justification For Why Not Lower Score:**

N/A

---

### Decision · Program_Chairs · 2024-01-16

Reject